# GREmLN: A Cellular Graph Structure Aware Transcriptomics Foundation Model

## Abstract

The ever-increasing availability of large-scale single-cell profiles presents an opportunity to develop foundation models to capture cell properties and behavior. However, standard language models such as transformers benefits from sequentially structured data with well defined absolute or relative positional relationships, while single cell RNA data have orderless gene features. Molecular-interaction graphs, such as gene regulatory networks (GRN) or protein-protein interaction (PPI) networks, offer graph structure-based models that effectively encode both non-local gene token dependencies, as well as potential causal relationships. We introduce GREmLN (**G**ene **R**egulatory **Em**bedding-based **L**arge **N**eural model), a foundation model that leverages graph signal processing to embed gene token graph structure directly within its attention mechanism, producing biologically informed single cell specific gene embeddings. Our model faithfully captures transcriptomics landscapes and achieves superior performance relative to state-of-the-art baselines on cell type annotation, graph structure understanding, and fine-tuned reverse perturbation prediction tasks. It offers a unified and interpretable framework for learning high-capacity foundational representations that capture complex, long-range regulatory dependencies from high-dimensional single-cell transcriptomic data. Moreover, the incorporation of graph-structured inductive biases enables more parameter-efficient architectures and accelerates training convergence.

## 1 Introduction

Interaction networks have been broadly used to represent the molecular logic that mediates the effect of genetic and epigenetic events in determining cell states and whose dysregulation is responsible for a variety of human diseases, including cancer(Califano & Alvarez (2017); Chen et al. (2014)). We propose that advances in multi-omics single cell profiling, such as single cell RNA-seq (scRNA-seq), coupled with cellular graphs, provide a unique opportunity to create foundation models that improve our ability to elucidate the mechanisms that underlie tumorigenesis, tumor plasticity, and drug responses. Recent attempts to train large-scale transformer-based foundation models with scRNA data demonstrated potential in terms of both cell-level tasks (e.g.: cell type prediction, batch correction) and gene-level tasks (e.g.: gene regulatory network analysis, perturbation response prediction)(Cui et al. (2024); Hao et al. (2024); Theodoris et al. (2023)). However, these models represent genes as discrete tokens and model cells as sequences, applying standard multi-head self-attention mechanisms, either in their original form (Theodoris et al. (2023)), with attention biases informed by gene-gene relationships (Cui et al. (2024)), or using randomized nonlinear kernel functions to encode structural priors (Hao et al. (2024)). These formulations overlook the fact that genes products in the cell, as represented in scRNA-seq profiles, lack inherent sequential order or positional semantics, which is beneficial for inducing stable long-range attention decay patterns that facilitate the Transformer's capacity to model dependencies and generalize to sequences beyond those encountered during training(Kazemnejad et al. (2023); Su et al. (2021); Ke et al. (2020)).

In this work, we introduce GREmLN(**G**ene **R**egulatory **Em**bedding-based **L**arge **N**eural model), a scRNA foundation model that leverages gene token graph structure to encode biologically meaningful relative position information and long-range dependency into single cell level gene embeddings. We first detail the formulation of our attention mechanism based on graph signal processing and numerical analysis. By applying a diffusion kernel to the graph Laplacian, we construct a kernel Gram

matrix that can be used to transform the query embeddings, thereby enabling the self-attention mechanism to be constrained and thus structured by the underlying graph. The diffusion process itself further helps capture long-range dependencies between gene nodes in the network. We also implement a Chebyshev polynomial based approximation of the kernel Gram matrix to scale to large graphs and long gene sequences. We then evaluate model performance with both cell level and gene level tasks such as cell type annotation with cell embeddings and graph structure understanding of gene embeddings to demonstrate superior performances over scRNA foundation model baselines.

## 2 RELATED WORK

**Graph Transformers** A growing body of work integrates graph structural priors into Transformer architectures to enhance relational reasoning and introduce domain-specific inductive biases Min et al. (2022a). These methods generally fall into two categories: (1) incorporating graph structure into positional embeddings, and (2) modifying attention computations using graph information.

Popular positional encoding schemes include classical Laplacian eigenvectors Dwivedi & Bresson. (2020) and their permutation-equivariant variants Lim et al.; Huang et al.. Other approaches include SVD-derived embeddings Hussain et al. (2021), centrality and degree features Ying et al. (2021), and hop-based relative embeddings via auxiliary GNNs Zhang et al. (2020). However, such embeddings often scale poorly to large or high-order graphs and risk introducing noisy or biased priors, limiting generalization. The work in Kanatsoulis et al. (2025) recently proposed learnable, GNN-generated encoding that act as drop-in, computationally efficient alternatives for Laplacian-based methods.

For attention modification, methods include designing equivariant attention mechanisms Tholken & Fabritiis (2022), using ego-centric subgraphs Zhao et al. (2021), or appending GNN-based features with permutation-invariant Transformers Wu et al. (2021). Others use attention masking Mialon et al. (2021); Min et al. (2022b) or bias terms derived from shortest path distances or proximity metrics Ying et al. (2021); Zhao et al. (2021). Yet, hard masking blocks all information flow and impairs gradient stability, while additive biases only shift attention scores without enabling structural message passing.

In contrast, we embed graph structure into a spectral kernel that filters noise and injects multi-hop, spectrum-aware inductive bias at low cost. This enhances convergence, preserves gradient propagation across regulatory edges, and enables each query to integrate contextual information from biologically related neighbors.

**Biological Foundation Models** Similar to their use in natural language understanding, foundation models learn robust representations from biological data. Early work such as ESMAlexander Rives et al. (2021) or ProGenet al (2020) focused on proteins sequences but have since expanded to many other modalities such as RNA sequencesZou et al. (2024), small molecules Seyone Chithrananda & Ramsundar (2020), and most relevantly scRNA-seqCui et al. (2024),Theodoris et al. (2023). Foundation models for scRNA-seq frame the problem analogously to natural language, where each cell is a document and each gene is a word. All methods aim to develop a meaningful latent representation of RNA expression in a given cell that can subsequently be used for downstream tasks such as batch effect removal Cui et al. (2024), cell type classification Theodoris et al. (2023), drug sensitivity predictionHao et al. (2024), or the effects of genetic perturbationsTheodoris et al. (2023).

One key difference of scRNA-seq to other modalities is that scRNA-seq represents the expression value of each gene as a continous rather than discrete variable. Most models solve this by simply discretzing expression into bins and modeling expression using standard auto-regressive Cui et al. (2024) or masked-token modelingTheodoris et al. (2023) objectives. Newer models such as sc-Foundation model the read depth of each cell and employ a masked regression objective to better model the continuous nature of RNA expression Hao et al. (2024). Another challenge with modeling scRNA-seq is the inherent lack of sequential structure. scRNA-seq expression vectors are unordered sets, and so standard sequential based positional encodings cannot be applied. Existing methods simply fix an arbitrary gene order; thus, this challenge remains unsolved.

## 3 MODEL ARCHITECTURE

We propose a new model architecture that integrates gene token graph structure to facilitate the learning of single-cell-specific gene embeddings. By employing this approach, the model gains the ability to capture meaningful long-range dependencies among gene tokens, while also enriching the feature space with relevant positional and structural relationships between these tokens. Such a framework applies to arbitrary gene token interaction graphs, including gene regulatory networks, protein-protein interaction networks and protein subcellular localization networks.

### 3.1 TOKENIZATION AND INITIAL EMBEDDINGS

We can extract two types of information from a scRNA count matrix: (1) Gene identity, and (2) Gene expression value, which is a measurement of how expressed a gene is in a cell. An unprocessed scRNA count matrix (raw count) contains positive integers representing the number of RNA transcript copies of a gene. The inputs to the model are a gene identity embedding matrix (denoted as $E^g$), and a gene rank embedding matrix (denoted as $E^r$). We construct the gene identity embedding for cell $i$ as:

$$E^g = emb_g(id(g_1^i),\ id(g_2^i),\ \dots,\ id(g_G^i))$$

where $emb_g$ is a learnable embedding layer. Gene IDs are never masked. To construct the gene rank embedding, we first compute the rank of gene $j$ in cell $i$ as:

$$r(g_j^i) = k, \hat{X}_{ij} \in [b_k, b_{k+1}]$$

where $\hat{X}$ is the normalized expression matrix, and $b_k, b_{k+1}$ are lower and upper bounds of bin $k$. The normalized expression matrix is obtained by normalizing the total counts of all cells to sum up to a fixed number (in our case $10^6$) and then log-plus-one transforming it. The <MASK> rank token is randomly assigned to 30% of the rank values. The gene rank embedding of cell $i$ is then given by:

$$E^r = emb_r(r(g_1^i),\ r(g_2^i),\ \dots,\ r(g_G^i))$$

where $emb_r$ is a learnable embedding layer. The input embedding to the transformer is then given by the concatenation of two representations:

$$E = [E^g || E^r]$$

We also prepend a <CLS> token to the beginning of each cell sentence that attends to all genes. The <CLS> token is not part of the graph and is not involved in any graph operations.

### 3.2 GRAPH DIFFUSION KERNEL ATTENTION

Leveraging the fact that we observe existing token graphs, we incorporate graph-based structural information into the attention mechanism by transforming each query embedding with a diffusion kernel Gram matrix derived from the token graph. In doing so, the query vectors become conditioned on graph topology—effectively biasing the attention process to reflect the gene-gene relationships encoded in the graph. Key and value embeddings, meanwhile, are not transformed to preserve the original information in token embeddings. We aim to develop general embeddings that inherently capture token graph structures. More specifically, assume the token graph adjacency matrix is $A$, we compute the normalized Laplacian matrix as:

$$L = I - D^{-1/2}AD^{-1/2},$$

where $D$ is the diagonal degree matrix. Using the normalized Laplacian matrix ensures that the eigenvalues remain bounded when mapping the graph to the spectral domain. To that end, let $L \in \mathbb{R}^{G \times G}$ be a symmetric matrix with spectral decomposition

$$L = U \Lambda U^\top, \qquad U = [u_1, \dots, u_G], \Lambda = \text{diag}(\lambda_1, \dots, \lambda_G).$$

For a non–negative spectral filter $\kappa : \mathbb{R} \to \mathbb{R}_{\geq 0}$ we define the *kernel Gram matrix*

$$\Phi_L = \sum_{i=1}^{G} \kappa(\lambda_i)\, u_i u_i^\top = U\, \kappa(\Lambda)\, U^\top$$

with $\kappa(\Lambda) = \mathrm{diag}\big(\kappa(\lambda_1), \ldots, \kappa(\lambda_G)\big)$. Because $\kappa(\lambda_i) \geq 0$ for every $i$, the matrix $\Phi_L$ is symmetric positive semidefinite and therefore constitutes the matrix representation of an orthogonal projection operator onto the Reproducing Kernel Hilbert Space $\mathcal{H}_\kappa$ whose reproducing kernel satisfies

$$k_\kappa(i,j) \;=\; e_i^\top \Phi_L e_j, \qquad i, j \in \{1, \ldots, G\}.$$

where $\kappa(.,.)$ is any kernel function.

$$\Phi_L(Q) = U \exp(-\beta\Lambda) U^T Q$$

We then compute dot-product attention with the diffused query embedding as:

$$Attn(Q, K, V, L) = \mathrm{softmax}\Big(\frac{\Phi_L(Q) K^T}{\sqrt{d}}\Big) V$$

where un-softmaxed attention score for each pair is given by the following similarity measure:

$$S_{ij} = q_i^\top \Phi_L^{1/2} k_j$$

This bilinear score incurs asymmetric smoothing which steers the attention mechanism toward interactions that respect the graph-implied low-frequency (long-range) decay, while still allowing keys to contribute high-frequency detail.

## 3.3 TRAINING OBJECTIVES

We formulate the pre-training objective as a masked modeling problem conditioned on graphs. Assume we have $G$ genes, let $X^r = (X_1^r, X_2^r, .., X_G^r)$ be the bin values of genes in a cell, $L^c$ be the normalized Laplacian matrix of the token graph specific to cell type $c$ which the cell belongs to, $m = (m_1, m_2, ..., m_G)$ be a random binary mask vector where $m = 1$ represent genes being masked. The likelihood of all masked genes is given by

$$p_\theta(X_m^r | L^c, X_{\bar{m}}^r, E^g) = \prod_{i, \forall m_i = 1} p_\theta(x_i^r | L^c, X_{\bar{m}}^r, E^g)$$

where $E^g$ is the gene identity embedding, $X_m^r$ are the bin values of masked genes, $\theta$ are all learnable parameters, and $X_{\bar{m}}^r$ are the bin values of unmasked genes. Denote the transformer encoder with graph diffusion kernel attention as $enc$ and the linear decoder network as $dec$, a masked gene $i$'s conditional distribution can be parameterized as:

$$E_{\bar{m}}^r = emb_r(X_{\bar{m}}^r)$$
$$H = enc(L^c, E_{\bar{m}}^r, E^g)$$
$$p_\theta(x_i^r | L^c, X_{\bar{m}}^r, E^g) = dec(H_i) = \sigma(W H_i + b)$$

where $H$ is the learned embedding by the transformer encoder. The pre-training objective is then given by maximizing the expected conditional log likelihood:

$$\mathcal{L} = -\mathbb{E}_{X^r \sim \mathcal{D}} \mathbb{E}_{m \sim p(m)} \Big[ \sum_{i, \forall m_i = 1} \log p_\theta(x_i^r | L^c, X_{\bar{m}}^r, E^g) \Big]$$

where $\mathcal{D}$ is some ground truth data distribution and $p(m)$ is the sampling distribution for the mask.

## 3.4 APPROXIMATION OF LARGE KERNEL GRAM MATRICES

Due to the size of biological networks and length of cell sentences, performing operations such as matrix exponential and spectral decomposition every batch to transform query embeddings becomes computationally expensive and difficult to scale. To alleviate this issue, we take advantage of Chebyshev polynomials to approximate the kernel Gram matrix. More specifically, if a token graph has $G$ nodes, we approximate the transformed query embedding with Chebyshev polynomial truncated at term $K \ll G$ as:

$$\Phi_L(Q) = U \exp(-\beta\Lambda) U^T Q$$

$$\approx \sum_{k=1}^{K} c_k T_k(L) Q$$

to efficiently compute the kernel gram matrix of large graphs. This approximation can be solved with Gauss-Chebyshev quadrature, which we detail in Appendix A.3. We also provide an analysis discussing the asymptotic bound of runtime for this operation in Appendix A.4.

## 4 Validation Experiments

In this section we compare our model to state-of-the-art transformer-based single cell transcriptomics foundation models, namely scGPTCui et al. (2024), GeneformerTheodoris et al. (2023), and scFoundationHao et al. (2024). For our graph input, we use gene regulatory networks and PPI networks as examples due to their broad availability, but other types of gene token graphs can be used as well. The detail for network generation is in Appendix A.1 and A.9. Additionally, we employ evaluation models with minimal representational capacity—such as linear models and shallow MLPs—to ensure that performance predominantly reflects the quality of the pre-trained embeddings rather than the expressiveness of the downstream model. First, we assess the model's performance on gene expression reconstruction and cell type classification to evaluate the quality of cell embeddings. Subsequently, we evaluate the extent to which the learned gene embeddings capture and encode underlying graph structural information. Lastly, we evaluate the model's ability to predict perturbation labels of cells from single target Perturb-Seq data, both from frozen and fine-tuned embeddings.

### 4.1 Evaluation datasets

We select evaluation datasets that are non-overlapping with those used during pre-training to assess the generalizability of all models. In particular, we focus on a dataset consisting of healthy human immune cells, as we are interested in downstream applications of our model to human immune cell state engineering. Aside from healthy immune cells, we also test the model on diverse datasets derived from cancer-infiltrating immune cells and non-immune cells. For perturbation tasks, we use the dataset from Adamson et al. (2016). Details and availability of the datasets can be found in Appendix A.8.

### 4.2 Transcriptomic Landscape Learning and Cell Type Annotation

We first validate of our model's effective learning of mRNA transcripts by training a simple linear regression model to reconstruct gene expression from frozen pre-trained embeddings. Our model is able to accurately reconstruct gene expression for both healthy and diseased human cells(Appendix A.6). We then assess the quality of our learned embeddings' cellular representations by measuring their ability to recapitulate cell type.

Since our model requires graphs as input, we only allow the model to see integrated networks that combines all cell-type specific GRNs to avoid inadvertently leaking cell-type labels.

**Bayesian Graph Integration**   Let $X_{\text{train}}$ be the training gene–expression matrix and let $X_{\text{test}}^{(i)}$ denote the profile of a single test cell $i$. For every cell type $c \in \mathcal{C}$ we compute the posterior edge-exclusion probability $\Pr\big(e \notin \mathcal{G}_c \mid X_{\text{train}}, C = c\big)$ and declare the edge present in the type-specific estimate $\widehat{\mathcal{G}}_c$ whenever this posterior does not exceed a user-defined false-edge level $\alpha$,

$$e \in \widehat{\mathcal{G}}_c \iff \Pr\big(e \notin \mathcal{G}_c \mid X_{\text{train}}, C = c\big) \leq \alpha. \tag{1}$$

A $k$-nearest-neighbor classifier fitted exclusively on $X_{\text{train}}$ yields an empirical posterior $w_c^{(i)} = \Pr\big(C = c \mid X_{\text{test}}^{(i)}\big)$, with $\sum_{c \in \mathcal{C}} w_c^{(i)} = 1$. Using the law of total probability,

$$p_i(e) = \sum_{c \in \mathcal{C}} w_c^{(i)} \, \Pr\big(e \notin \mathcal{G}_c \mid X_{\text{train}}, C = c\big). \tag{2}$$

The regulatory graph supplied to the model for cell $i$ is finally obtained via the same $\alpha$-level threshold,

$$\widehat{\mathcal{G}}_i = \big\{ e : p_i(e) \leq \alpha \big\}. \tag{3}$$

Equations equation 1–equation 3 collectively recast the procedure as two successive Bayesian updates: first, learning type-specific edge posteriors on $X_{\text{train}}$; second, averaging these posteriors for each unseen cell under the $k$-NN–derived type posterior, thereby integrating multiple cell-type-specific graphs without ever exposing the true test-cell labels.

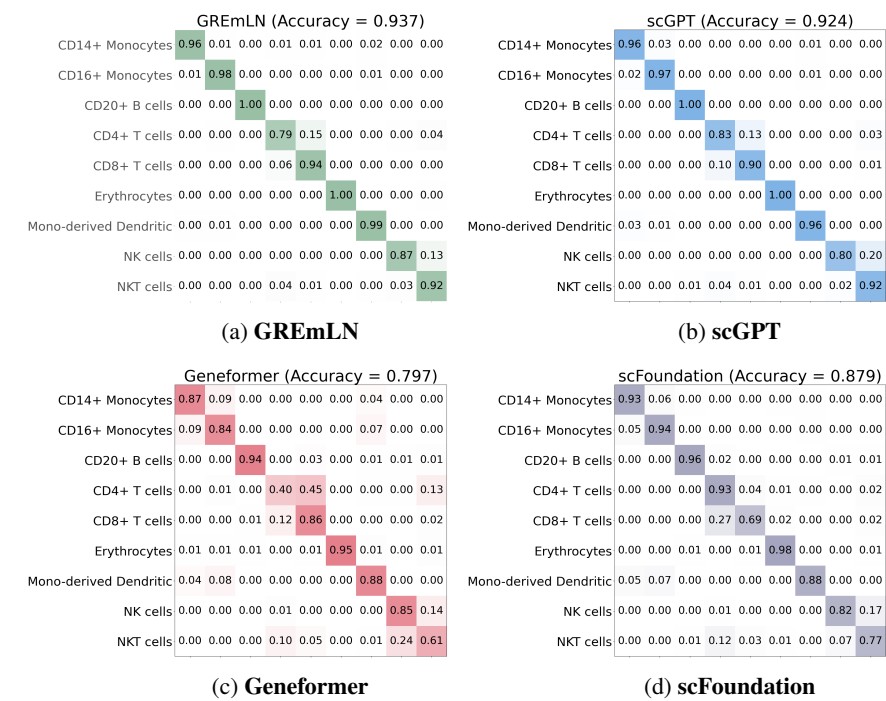

Figure 1: **Cell-type classification performance on the Human Immune dataset.** Shown are confusion matrices produced by (a) GREmLN, (b) scGPT, (c) Geneformer, and (d) scFoundation. Each heat-map cell uses the same colour scale so the three models can be compared directly.

| Dataset | Human Immune Cells | | | Non-Immune Cells(zero-shot) | | |
|---|---|---|---|---|---|---|
| Model | Precision | Recall | Macro F1 | Precision | Recall | Macro F1 |
| GREmLN | **0.941 ± 0.002** | **0.937 ± 0.003** | **0.939 ± 0.001** | **0.936 ± 0.002** | **0.936 ± 0.001** | **0.937 ± 0.001** |
| scGPT | 0.928 ± 0.003 | 0.924 ± 0.001 | 0.924 ± 0.002 | - | - | - |
| scFoundation | 0.887 ± 0.013 | 0.879 ± 0.017 | 0.879 ± 0.015 | 0.913 ± 0.002 | 0.904 ± 0.003 | 0.903 ± 0.003 |
| Geneformer | 0.800 ± 0.012 | 0.797 ± 0.010 | 0.792 ± 0.010 | 0.774 ± 0.015 | 0.768 ± 0.010 | 0.768 ± 0.012 |

Table 1: **Test cell type annotation performance across multiple tissue types.** CI is computed across 3 random initializations of test splits. scGPT is excluded from evaluation on the Held-out Non-Immune Cell dataset, as it does not constitute a zero-shot setting.

**Training cell type classifier** For each cell $i$, we begin with its embedding matrix $H_i \in \mathbb{R}^{L \times d}$, where $L$ is the gene dimension and $d$ is the embedding dimension. Next, we obtain a fixed-size cellular representation $h_i \in \mathbb{R}^d$ via mean pooling over the gene sequence dimension. Next, we fit a linear classifier to predict cell type given $h_i$ and the integrated graph $\widehat{\mathcal{G}}_i$ by minimizing cross-entropy loss between the estimated class distribution and the true cell type label. More specifically, assuming we have cell type label $y_i$, we aim to learn the distribution of cell type labels conditioning on the cell representation and the input graph. For cell type $c$ the conditional density is given by:

$$p(y_i = c|h_i, \widehat{\mathcal{G}}_i, f_\theta)$$

where $f_\theta$ are the classifiers being trained. We learn $f_\theta$ by optimizing cross entropy loss as:

$$\mathcal{L}(\theta) = -\sum_{c \in C} \sum_{i=1}^{N} \log p(y_i = c|h_i, \widehat{\mathcal{G}}_i, f_\theta)$$

Our model achieved highest performances on all classification metrics on the human immune cells dataset(Figure 1, Table 1). Additionally, for held-out non-immune cells we test the model's ability to perform zero-shot cell type annotation. Since scGPT are trained with the full CELLxGENE dataset, we exclude the model from this evaluation. We achieve superior zero-shot performances on the held-out non immune cell dataset(Table 1) for all metrics. We also point out that GREmLN

contains only 10.3 million learnable parameters (less than one third of all baselines' parameter sets and one tenth of scFoundation, see details in Appendix A.5), which highlights the predictive power of gene regulatory network guidance. Our key contribution lies in tailoring the attention mechanism to encode biological inductive bias, rather than increasing model size.

### 4.3 GRAPH STRUCTURE UNDERSTANDING

We assess the learned embeddings' capacity to capture and generalize graph structure by evaluating edge prediction on unseen gene regulatory networks from held-out cells. Specifically, we randomly mask a subset (15%) of edges and measure how well the model recovers them when conditioned on the learned embeddings and the incomplete graph. Let $H$ be learned gene embeddings from a model

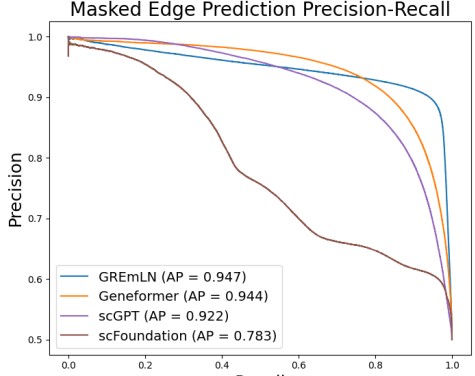

Figure 2: **Benchmarking performance for graph understanding on the Human Immune Dataset.** Left: Receiver operating characteristic curve with area under curve (AUROC) metrics. Confidence intervals computed with cross validation. Right: Precision-recall curve with average precision (AP) metrics.

and $A$ be the adjacency matrix of the graph, the objective is to learn the conditional probability of a masked edge existing, i.e.:

$$p_\theta(A_{i,j} = 1 | \tilde{A} = A \odot \mathcal{M}, H)$$

We use the learned embeddings $H$ from pre-trained models and parameterize the conditional likelihood with a model $f_\psi$ where $\psi$ are learned model parameters. More specifically, for an edge $(i, j)$:

$$p_\theta(A_{i,j} = 1 | \tilde{A} = A \odot \mathcal{M}, H) = f_\psi(i, j, \tilde{A}, H_i, H_j)$$

where $H_i, H_j$ are learned embeddings of nodes $i, j$ respectively. To learn the model $f_\psi$ we treat an edge-label pair $\{(i, j), y_{ij}\}$ as i.i.d samples from some data generating distributions. The label $y_{ij} = 1$ if the edge $(i, j)$ is a real masked edge and $y_{ij} = 0$ if the edge is a fake edge that does not exist in $A$. The training objective is then given by:

$$\mathcal{L}(\psi) = - \sum_{\{(i,j), y_{ij}\} \sim \mathcal{D}} [y_{ij} \log f_\psi(i, j, \tilde{A}, H_i, H_j) + (1 - y_{ij}) \log(1 - f_\psi(i, j, \tilde{A}, H_i, H_j))]$$

In practice, we freeze our encoder weights and incorporate the partially observed graph $\tilde{A}$ through the attention mechanism to condition our embeddings. For baseline models, we similarly condition on $\tilde{A}$ by passing the learned gene embeddings through a graph attention network using the same masked graph. We first benchmark the GRN structure understanding task on healthy human datasets to evaluate model performances with seen cell types but unseen data/graph. GREmLN achieves the best performance by a significant margin in terms of both ROC and PRC, while Geneformer is also competitive due to their tokenization and pre-trianing objective focusing on learning gene embeddings that are GRN-aware (Figure 2).

To further evaluate the model's capacity for structural generalization, we assessed performance on graph structure understanding tasks involving previously unseen cell types. Given that pathological

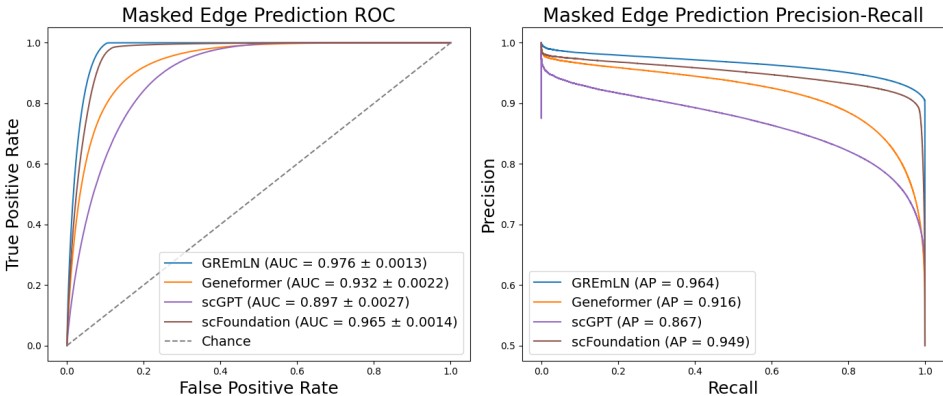

Figure 3: **Benchmarking performance for graph understanding on the Cancer Infiltrating Myeloid dataset.** Left: Receiver operating characteristic curve with area under curve (AUROC) metrics. Confidence intervals computed with cross validation. Right: Precision-recall curve with average precision (AP) metrics.

conditions such as cancer often induce local rewiring of gene regulatory networks, we specifically tested the model on cancer-infiltrating myeloid cells from distinct tissue origins. Evaluation was conducted on held-out cancer types to simulate out-of-distribution generalization. GREmLN consistently outperformed all baseline models across all performance metrics (Figure 3), demonstrating superior generalization to previously unseen regulatory graphs within complex and heterogeneous biological systems.

## 4.4 GRAPH ABLATION

Using gene regulatory networks as input to our model introduces risk of information duplication, as the graph itself is inferred from gene expression profiles. To clarify that gene regulatory networks indeed provide more information, we perform an ablation study comparing our model to a naive transformer with regular dot product attention.

| Task, Dataset, Metric | GREmLN | Vanilla(ablation) |
|---|---|---|
| Masked edge prediction, Human Immune, AUROC | 0.957 | 0.683 |
| Masked edge prediciton, Cancer myeloid, AUROC | 0.976 | 0.941 |
| Cell type annotation(zero-shot), Non-immune cells, F1 | 0.937 | 0.816 |
| Cell type annotation, Human-immune cells, F1 | 0.939 | 0.921 |

Table 2: **Ablation study results on all tasks.**

Graph ablation yields substantial performance degradation across tasks as shown in Table 2, with the largest declines on (i) masked-edge prediction in the Human Immune cohort and (ii) zero-shot cell-type annotation in the held-out Non-Immune cohort. Both settings exhibit significant testing data distribution shift: the Human Immune test split includes previously unseen cell types with markedly different underlying GRNs, and the Non-Immune dataset constitutes a fully out-of-domain evaluation. Because the GRN-aware GDK acts as an inductive bias for structure-aware representation learning, its removal diminishes out-of-distribution generalization. By contrast, edge prediction in the Cancer Infiltrating Myeloid cohort and annotation within Human Immune cells are less affected: Myeloid cells share lineage and thus have closely related GRNs, while major Human Immune cell types are well separated in expression space, making these tasks less reliant on the structural prior.

## 4.5 FINE-TUNING FOR REVERSE PERTURBATION PREDICTION

We lastly evaluate fine-tuning for reverse perturbation prediction on Perturb-Seq data: given a perturbed cell expression profile, the model needs to infer the corresponding perturbation label. Single-

gene CRISPR perturbations induce only subtle transcriptomic shifts, yielding low class separability and making the task intrinsically challenging. For a fair assessment, we first benchmark GREmLN with scFoundation and Geneformer, which report state-of-the-art performance, using frozen embeddings. We also include a linear model as a baseline. We then fine-tune GREmLN with classification entropy loss and report GREmLN's fine-tuned performance. We didn't fine-tune the baseline models due to time and computational resource constraints. To further emphasize the generalizability of our graph-structure based attention mechanism, we train GREmLN with both GRN and curated PPI network inputs and fine-tuned with both. See Appendix A.9 for PPI network details.

| Model | Accuracy | F1 Score |
|---|---|---|
| scFoundation-frozen | 0.219 | 0.209 |
| Geneformer-frozen | 0.244 | 0.185 |
| Lasso logistic regression | 0.281 | 0.127 |
| GREmLN-GRN-frozen | 0.257 | 0.209 |
| GREmLN-PPI-frozen | **0.304** | **0.275** |
| GREmLN-GRN-finetuned | **0.418** | **0.380** |
| GREmLN-PPI-finetuned | 0.377 | 0.320 |

Table 3: **Reverse Perturbation Prediction Performance Across Models**. Best performances are bold-faced for frozen and fine-tuned embeddings

We see from table 3 that GREmLN achieve state-of-the art performance with and without fine-tuning. Notably, GREmLN with PPI priors performs better in the frozen-embedding setting, whereas GRN-based training yields superior fine-tuning. This matches a bias–variance trade-off: PPIs are uniform across cell types (stronger prior, lower variance), while GRNs are cell-type specific (more flexible, lower bias for adaptation).

## 5 CONCLUSION AND DISCUSSION

In this work we introduce GREmLN, a graph-aware foundation model that integrates gene token graphs, enabling the learning of biologically informed gene embeddings through graph-induced ordering. The learned embeddings can be used to reconstruct gene expression with a linear regression model, demonstrating that transcriptomics information is well captured. The model additionally achieves state-of-the-art performance in predicting held-out cell types and unseen regulatory structures, illustrating its ability to capture generalizable rules of gene regulation and representation of cell states. The model can be pre-trained with both GRN and PPI inputs, and can be fine-tuned to distinguish cells with small transcriptomics differences from Perturb-seq data with different perturbation labels.

While the functional search space for gene interactions is intractable even for large models, incorporating validated regulatory interactions significantly improves performance across downstream tasks Ideker & Krogan (2012); Bar-Joseph et al. (2003); Alvarez et al. (2016). Our architecture supports arbitrary molecular interaction graphs and leverages well-established methods for network inference Margolin & Nemenman (2006) alongside diffusion-based graph kernels Paull et al. (2013); Leiserson et al. (2015). Future directions include fine-tuning cell and gene embeddings for specific tasks—such as modeling combinatorial perturbations and identifying optimal interventions—while using attention-based interpretability to recover core regulatory modules.

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

# A APPENDIX

## A.1 GENE REGULATORY NETWORK CONSTRUCTION

To construct gene regulatory networks, we use the extensively validated ARACNe algorithmMargolin & Nemenman (2006) which infers the transcriptional targets of regulatory proteins (represented as edges in a network) by estimating the mutual information between their gene expression via bootstrapping and adaptive partitioning, followed by elimination of indirect targets via the data processing inequality theorem. ARACNe is a key component of several well-validated algorithms that have been effective in elucidating master regulator proteins representing mechanistic determinants of tumor cell states as well as in determining their sensitivity to small molecules, including use in clinical trialsAlvarez et al. (2016; 2019); Mundi PS (2023); Paull et al. (2013); Paull EO (2021); Zeleke et al. (2023), thus demonstrating the biologically meaningful nature of the inferred networks, which makes ARACNe networks a natural and potentially useful choice for implementing graph-enriched embeddings in foundation models.

Gene regulatory networks vary widely across cell types, so prior to model pre-training, we infer a unique GRN for each cell type in our training corpus. To reconstruct each network, we first run ARACNe within every Louvain-algorithm-inferred cluster. The resulting subnetworks are then consolidated into a single network by evaluating the likelihood of edge recurrence under a binomial null model, allowing the false discovery rate to be controlled at 5%. Forming a consensus network from cluster-specific subnetworks offers several advantages: (1) consensus networks are robust to potentially significant batch effects; (2) almost all cells of each cell type contribute to the final network, and (3) multiple possible cell states are faithfully represented in the single network, enabling embeddings to capture the full granularity of single-cell biological states.

## A.2 PREPROCESSING PIPELINE AND IMPLEMENTATION DETAILS

**Preprocessing Pipeline** Single cell RNA sequencing data is inherently noisy, subject to batch effects, sequencing errors, and high dropout rates. In fact, the sparsity of the expression profiles in our dataset is approximately 80%. Extracting a meaningful signal from such profiles is challenging; thus, it is critical to pre-process this data appropriately before feeding it to the model. In this work we choose to discretize the cell sentences. Discretization is not only necessary for the masked token prediction pre-training task but also to attenuate batch effects in scRNA-seq data. Indeed, rank-based binning produces a dataset-invariant scaling. Since we optimize cross entropy loss during pre-training, the optimal choice of number of bins would be enough bins to produce a near-uniform occupancy distribution. In practice, the bin number is treated as a hyper-parameter and selected via grid search on validation data with an increment of 50 bins. Previous single cell transcriptomics foundation models such as scGPT and Geneformer also rely on similar pre-processing steps.

Our pre-processing pipeline consists of the following steps: quality control and normalization, quantization, regulatory network generation, and caching. (1) The cells are controlled for quality by filtering out any cells not meeting our UMI sequencing depth requirements. For a given cell, the UMI count of a gene is a measure of that gene's expression level. UMI count thresholds are set to a minimum and maximum of 1K and 100K reads, respectively, and - if needed - are dynamically updated to keep only the middle 90% of cells by total UMI count. Cells with insufficient total UMI counts are removed as they likely represent faulty or empty reads from the sequencing process, and cells with exceedingly large total UMI counts are removed as they likely represent doublets and multiplets (the read contains information from more than one cell). The remaining expressions are log1p normalized to reduce the range of expression values and stabilize the expression variance. (2) After quality controlling, the cells are quantized into 100 expression bins. The zero-th bin is assigned to all zero-expression genes and the remaining expressed genes are assigned to their corresponding quantile for that cell; the bin number increasing with expression. This provides a discretized relative expression of the genes in each cell, preserving cell insights while reducing the computational burden of processing this data during training. (3) Gene Regulatory Networks (GRNs) are then generated for each cell-type using the ARACNe algorithmMargolin & Nemenman (2006), as described in 3.2. The use of GRNs is at the heart of the novelty of this model, framing expression signatures in a regulatory context. (4) In the final caching step, a GRN sub-network including only the non-zero expressed genes is created for each cell. It is also required that each single cell meets the minimum number of 500 expressed genes.

To evaluate the model's zero-shot capabilities, 15% of cell-types, along with their GRNs, are held out as the unseen-graph validation set. The cells of the remaining 85% are randomly split into the seen-graph validation set and the training set. The seen-graph validation set accounts for approximately 15% of total cells, leaving the last 70% for the training set.

**Implementation Details** The model is implemented using PyTorchAdam et al. (2019) and FlashattentionTri et al. (2022). Optimization are performed with the AdamW optimizer in PyTorch for both pre-training and validation models. Validation-based early stopping are implemented for all downstream regressors/classifiers that takes in frozen embeddings. The architecture of downstream models for all validation experiments are identical across the different foundation models within the same task.

## A.3 CHEBYSHEV APPROXIMATION OF DIFFUSED QUERY EMBEDDINGS

We first compute the Chebyshev coefficients $c_k$. Since our Laplacian matrix is normalized, the range of eigen values is $[0, 2]$. Since Chebyshev polynomials are orthogonal on $[-1, 1]$, we first map to this domain by setting

$$x = \frac{2\lambda}{\lambda^*} - 1$$

where $\lambda^*$ is the max eigenvalue, which means $\lambda^* = 2$. Solving for $\lambda$ yields:

$$\lambda = \frac{\lambda^*}{2}(x + 1) = (x + 1)$$

We can then compute the Chebyshev coefficients numerically with Gauss-Chebyshev quadrature. The Chebyshev coefficients are given by:

$$c_k = \frac{2}{\pi} \int_{-1}^{1} \frac{f(\lambda)T_k(x)}{\sqrt{1 - x^2}} dx \tag{4}$$

$$= \frac{2}{\pi} \int_{-1}^{1} \frac{\exp(-\beta\lambda)T_k(x)}{\sqrt{1 - x^2}} dx \tag{5}$$

$$= \frac{2}{\pi} \int_{-1}^{1} \frac{\exp(-\beta(x + 1))T_k(x)}{\sqrt{1 - x^2}} dx \tag{6}$$

To compute this integral numerically, the non-angular quadrature points $x_i$ can be converted to a function of angles $\theta$ as:

$$x_i = \cos\theta_i = \cos(\frac{\pi(i - \frac{1}{2})}{N})$$

where $N$ are number of total quadrature points. Notice that $T_k(x) = T_k(\cos(\theta)) = \cos(k\theta)$. We differentiate both sides to obtain:

$$\frac{dx}{d\theta} = -\sin\theta \rightarrow dx = -\sin\theta d\theta$$

Denote $g(x) = \exp(-\beta(x + 1))$, substitute this change-of-variables form into the original integral yields:

$$c_k = \frac{2}{\pi} \int_{-1}^{1} T_k(x)\frac{g(x)}{\sqrt{1 - x^2}} dx \tag{7}$$

$$= \frac{2}{\pi} \int_{0}^{\pi} \cos(k\theta)\frac{g(\cos\theta)}{\sin\theta} * \sin\theta d\theta \tag{8}$$

$$= \frac{2}{\pi} \int_{0}^{\pi} \cos(k\theta)g(\cos\theta)d\theta \tag{9}$$

Notice that in (5) the integral range after variable change becomes $[\pi, 0]$ and we flip the range because of the negative sign from the change-of-variables form. The resulting integral after change-of-variables can be approximated as:

$$c_k \approx \frac{2}{N} \sum_{i=1}^{N} g(\cos\theta_i) \cos(k\theta_i)$$

which can be efficiently computed. The Chebyshev polynomial terms and the subsequent transform of the query embedding $Q$ is given by:

$$T_0(L)Q = Q$$
$$T_1(L)Q = LQ$$
$$T_k(L)Q = 2LT_{k-1}(L)Q - T_{k-2}(L)Q$$

### A.4 RUNTIME ANALYSIS

Here we provide run time analysis of both our pre-processing steps and our attention computation.

**GRN inference**  With respect to ARACNe's scalability and resource requirement, we can clarify that the algorithm requires minimal compute resources, even when considering hundreds to thousands of samples. Importantly, since network generation is a post-clustering step and ARACNe requires tissue specific consistency, there is no need to use more than a small subset of sample—typically 500 profiles—regardless of cluster or dataset size. As there are a limited number of stable and transitory biological states physiologically possible, the computational cost will scale far below linear when approaching billions of cells, and we will update the text to clarify this critical point. Additionally, network generation is a pre-processing step so it's performed once all together before model training.

**Attention computation**  As for scalability of our model, our model's runtime is bounded by the runtime complexity of a vanilla attention model. Assume we have $G$ tokens, hidden size $d$, $N$ quadrature points, and $K$ terms of Chebyshev recursion, the runtime for performing the operation of estimating Gram matrix and transforming the query is given by

$$\mathcal{O}(NK) + \mathcal{O}(K \cdot G\delta \cdot d)$$

where $\delta$ is the average degree. The first term is the runtime of computing the chebyshev coefficients(see appendix A.3), while the second part is the runtime of computing the sparse matrix multiplication of transforming query embeddings. Since biological networks are very sparse in general, we have $\delta, K \ll G$. Hence, our attention computation is asymptotically bounded by the runtime of a regular dot-product attention model with runtime $\mathcal{O}(G^2 d)$

### A.5 HYPERPARAMETER SELECTION AND TRAINING DETAILS

All hyperparameters and their values used to train the model are listed in the following table. The model tested contains 10.3 million parameters.

| Hyperparameters | Value |
|---|---|
| Transformer model dimension | 512 |
| Number of Transformer blocks | 3 |
| Number of attention heads per block | 8 |
| Activation | GeLU |
| Peak learning rate | 3e-4 |
| Batch size | 64 |
| Chebyshev truncation index($K$) | 64 |
| Diffusion rate($\beta$) | 0.1 |
| Number of quadrature points($N$) | 100 |
| Graph integration false edge threshold($\alpha$) | 0.05 |

We apply dynamic learning rate scheme to ensure scalable learning behavior. At the start of pre-training, we first apply a linear ramping of learning rate with start factor 1e-5 that lasts for 10% of the total steps. After peak learning rate is achieved, we apply cosine annealing for the rest of the training steps to promote generalization and avoid forgetting. The model was pre-trained for 1 full epoch on 2 Nvidia H100 GPUs in parallel. For the diffusion rate, larger betas lead to models capturing global patterns(longer diffusions), while smaller betas leads to local patterns(shorter diffusions). A good default value is $\frac{1}{\lambda_{min}}$ where $\lambda_{min}$ is the minimum non-zero eigenvalue of the Laplacian matrix. For the truncation order $K$, a good default value is close to the rank of the Laplacian matrix. In our specific case, we selected parameter values by grid-search around the default values.

## A.6 LINEAR PROBING OF GENE EXPRESSION

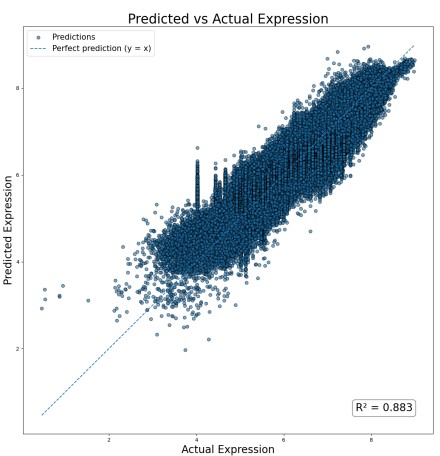 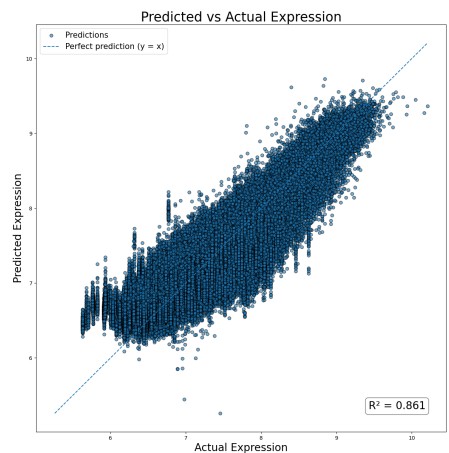

(a) **Human Immune Cells,** $R^2 = 0.883$      (b) **Cancer Infiltrating Myeloid,** $R^2 = 0.861$

Figure 4: **Reconstructing gene expression** Shown are predicted and ground truth gene expression from the learned GREmLN embedding. A linear regressor was trained on GREmLN embedding to predict gene expression.

## A.7 BASELINE MODEL CHECKPOINTS

Here we list the exact pre-trained checkpoint for the models used in the experiments:

**scGPT** We used the 33M parameter model pre-trained on normal human cells, which is available at `https://github.com/bowang-lab/scGPT`

**Geneformer** We used the 30M parameter model with version number GF-6L-30M-i2048, which is available at `https://huggingface.co/ctheodoris/Geneformer`

**scFoundation** We use the 100M parameter model available at `https://github.com/biomap-research/scFoundation`

## A.8 DETAILS OF VALIDATION DATASETS

**Human Immune Cells** This dataset includes cells from both human bone marrow and peripheral blood, covering 16 distinct immune cell types. We use the version provided by scGPT (`https://figshare.com/ndownloader/files/25717328`), with original cell type annotations from Luecken et alLuecken et al. (2021). This dataset serves as our primary benchmark for evaluating the model's ability to capture gene regulatory graph structure and to learn meaningful cell embeddings. We apply the same preprocessing steps as those used for the CELLxGENE corpus for model pre-training.

**Cancer Infiltrating Myeloid** This dataset comprises tumor-infiltrating myeloid cells (TMIs) from nine different human cancer types. TMIs are key regulators of tumor progression, with properties and functions that vary across cancer types. Similar to scGPT, we designate TMIs from six cancer types as a reference set for training and use the remaining types for out-of-distribution inference (query set). Pre-processing steps are identical to those applied to the CELLxGENE corpus. The dataset is available at: `https://drive.google.com/drive/folders/1VbpApQufZq8efFGakW3y8QDDpY9MBoDS`

**Held Out Non-Immune Cells** This dataset consists of 10 non-immune cell types that our model does not see during pre-training and validation, sourced from the CELLxGENE corpus (available

at: `https://cellxgene.cziscience.com/collections`) with the same pre-processing steps as the training dataset. The selected cell types are retinal rod, epithelial, perivascular, amacrine, respiratory basal, smooth muscle, neural progenitor, secretory, myofibroblast, and megakaryocyte, testing the model's ability to perform zero-shot annotation across distinct non-immune cell types.

**Adamson et. al. Perturb-seq** This dataset consists of cells from the K562 leukemia cell line perturbed by Perturb-seq. In total, there are 87 unique single-gene perturbations. Since many genetic perturbations have an almost undetectable impact on the transcriptional states of cells, we select the 20 perturbations that produce the greatest distributional shift in transcriptional states relative to a control cell population, as measured by centroid distance in PCA space. The resulting pared-down dataset used for reverse perturbation modeling thus contains 9,416 cells across 20 perturbation conditions. The dataset is publicly available at: `https://dataverse.harvard.edu/api/access/datafile/6154417`

## A.9 PPI NETWORK

We obtained our PPI network from HINT(High Quality Interactomes), which is avaliable at: `https://hint.yulab.org/`. HINT is a curated aggregate of PPIs from eight sources (BioGRID, MINT, iRefWeb, DIP, IntAct, HPRD, MIPS, PDB). It applies automated and manual QC to remove low-quality/erroneous interactions, updates nightly, and supports both interactive queries and bulk downloads. We also tried pre-train with PPI networks from STRING-DB(`https://string-db.org/`) with high edge confidence cutoffs. However, we observe empirically that HINT networks significantly outperforms STRING-DB networks. It is outside the scope of this work to compare and investigate PPI network qualities.

