# OpenReview forum: "GREmLN: A Cellular Graph Structure Aware Transcriptomics Foundation Model"
_ICLR.cc/2026/Conference — Submitted to ICLR 2026_

### Official Review · Reviewer_Dz2X · 2025-10-21

**Soundness:** 2
**Presentation:** 3
**Contribution:** 2
**Rating:** 2
**Confidence:** 4

**Summary:**

This paper addresses an important problem in the domain of foundation models for single-cell RNA sequencing (scRNA-seq). The authors propose GREmLN, a single-cell foundation model that incorporates cell-type-specific gene regulatory networks (GRNs) inferred directly from gene expression data. These GRNs inform relationships between genes through the graph Laplacian, offering biologically meaningful inductive biases. The model is evaluated across multiple downstream tasks—including cell type annotation, missing edge prediction, and reverse perturbation prediction—where GREmLN outperforms existing single-cell foundation models such as scGPT.

**Strengths:**

1. Well-motivated: Incorporating biologically meaningful structures such as cell-type-specific GRNs into foundation models is an important and timely idea in scFMs.
2. Clear presentation: The paper is generally easy to follow, although some methodological details would benefit from further clarification.
3. Comprehensive evaluation: The proposed approach is benchmarked across multiple tasks.

**Weaknesses:**

The key limitation is the reliance on an inferred GRN, because GRN inference is itself a challenging problem, particularly for large-scale settings with 20k genes. Does the accuracy of GREmLN depend heavily on the chosen GRN inference method? The accuracy of GRN inference can be substantial different across algorithms and datasets. Furthermore, several methodological and experimental details are insufficiently described. Please see my questions below.

**Questions:**

1. Chebyshev Polynomial Approximation: The use of Chebyshev polynomials to approximate the kernel Gram matrix (Lines 210–213) is motivated by the claim that “matrix exponential and spectral decomposition every batch … is computationally expensive and difficult to scale.” However, spectral decomposition of the fixed graph Laplacian need only be computed once, as the graph is not change during training. Did I miss something here?

2. Details of GRN inference: Was the GRN inferred over all 20k genes? How accurate is the inferred GRN? How sensitive is GREmLN’s performance to the specific GRN inference algorithm used? Different algorithms (e.g., GRNBoost2, DeepSEM, etc.) can yield markedly different GRNs. Recent benchmarks [1] indicate performance close to random for many inference methods in certain cell lines—would the authors consider presenting results across multiple GRN inference approaches to assess robustness?

3. GRN at the test time: The “Bayesian Graph Integration” step entails training a cell-type classifier (e.g., kNN on the expression matrix) to identify the cell type for a test datapoint, and using the averaged GRN. What is the accuracy of this classifier? How does misclassification at this step affect downstream performance? Using kNN on gene expression can hardly give an accurate prediction.

4. Edge prediction evaluation bias: The proposed evaluation masks a subset of edges in the GRNs and attempts to recover them.
Prior work [2] shows this setup can be biased—prediction accuracy can be spuriously high using only unmasked edges without leveraging any gene information, due to structural imbalances in gene connectivity. A more unbiased design would mask all edges related to specific genes (making those genes entirely unseen during training) and evaluate recovery.

5. Error bars or confidence intervals should be reported alongside mean performance metrics to convey statistical reliability.

6. Did the author use binned gene expression or the rank of gene expression values?

[1] Benchmarking algorithms for gene regulatory network inference from single-cell transcriptomic data, Nature Methods, 2020

[2] InfoSEM: A Deep Generative Model with Informative Priors for Gene Regulatory Network Inference, ICML, 2025

---

> ### Author Response · Authors · 2025-11-21
> **Rebuttal**
>
> We thank the reviewer for their insightful comments. We offer the following points to kickstart a discussion:
>
> 1. Limitations of GRN input: We thank the reviewer for this important comment. The graph diffusion–kernel attention framework we propose is a generic mechanism for injecting relational structure into attention scores. It requires only (i) a graph encoding pairwise relationships between tokens and (ii) informative token embeddings; it is not specific to gene-regulatory networks. In particular, the same construction applies to protein–protein interaction (PPI) graphs and protein subcellular localization networks, many of which are experimentally curated rather than computationally inferred (see Appendix A9 for our PPI input). As long as the underlying graph admits a positive semi-definite Laplacian, the associated diffusion kernel and its Chebyshev approximation define a valid Gram matrix, and the resulting query transformation remains well posed. Empirically, we demonstrate in Section 4.5 that using a PPI graph as input yields performance that exceeds the GRN-based variant, underscoring that the method is agnostic to the specific molecular network as long as it is biologically meaningful and sparse. We are currently extending this to additional classes of relational inputs. That said, when a GRN is used as the graph prior, the fidelity of the inferred GRN naturally influences embedding quality—precisely as one would expect from an inductive bias.
>
> 2. Chebyshev computation: The reviewer is completely correct to inquire on this. In practice, the kernel gram matrices for pre-training are precomputed and the same matrix can be used across layers of the model. This leads to a significant speed-up in pre-training (5x compared to on-the-fly computation). Chebyshev is proposed to account for runtime problems even when computing a single graph, since biological networks can get to around 5000, 10000, or even 20000 nodes. We will clarify this point and rewrite corresponding sections to make it more obvious.
>
> 3. ARACNe graphs are inferred from 4096 genes: including all transcription factors, completed with the most highly variable genes in that cell-type. At this time, we have also trained GREmLN on Protein-Protein Interaction (PPI) networks, which demonstrate good performance. No work has yet been conducted to determine the performance difference between GRN inference algorithms.
>
> 4. Error bars: We report error bars wherever feasible. However, repeated runs of large foundation models such as scFoundation, scGPT, and Geneformer are computationally expensive, which limits the number of independent replicates we can reasonably perform. We will include additional runs and corresponding uncertainty estimates where computational resources permit before the final submission deadline.
>
> 5. Expression binning: Expression values are binned into 100 expression bins. Thus, identical expression values are mapped to the same bin and similar expression values are sometimes mapped to the same bin if there are no free bin position left, in a "pigeonhole" fashion.
>
> 6. Bayesian Graph Integration Concern: The gene expression matrix is first projected onto the first 50 principal components of the training samples, after which  a kNN-classifier is used to classify test samples in this PCA space. On human immune cells, for example, this classifier achieves approximately 90% accuracy. Misclassification at this step naturally harms downstream classification performance, since an inaccurate cell type assignment leads to using a less precise graph that is not as biologically relevant and will transform the query vector in a non-biologically relevant way. Importantly, GREmLN achieves superior cell type classification performance while using imperfectly inferred GRNs, underscoring its robustness and generality.
>
> 7. Edge prediction evaluation bias: In our edge prediction evaluation, reported accuracies and AUC-ROCs are computed only on masked edges and false edges. During training, the model only sees unmasked edges, representing 85% of the original graph. During evaluation, only masked edges (the 15% of the graph not seen during training) and false edges are used for assessment. Therefore, unmasked edges cannot spuriously inflate prediction accuracy.

---

> > ### Comment · Reviewer_Dz2X · 2025-11-25
> > **Official comments**
> >
> > Thank you for your response which addressed some of my concerns! About the last point, evaluating performance on masked edges can still be biased if both nodes of a masked edge are seen during the training, because the model can simply use memorized node-specific information rather than learning generalizable representation to make prediction.

---

> ### Author Response · Authors · 2025-11-26
> **RE: Reviewer comments**
>
> We thank the reviewer for engaging in an active discussion and raising this important point. We agree that edge-level evaluation can, in principle, be biased if the model can rely purely on node-specific memorization. In our setup, two aspects mitigate this. First, for the graph-understanding task we freeze the pretrained encoder and train a lightweight edge-prediction head only on masked edges from training graphs; evaluation is then performed on masked edges from held-out GRNs (and, in the cancer setting, held-out cancer types). The predictor never receives supervision on the specific edges or graphs used at test time and only operates on continuous gene embeddings, not explicit node IDs, which restricts its ability to memorize individual edges. Second, the regime we care about in single-cell analysis is generalization across rewired graphs over a fixed gene set, not to unseen gene tokens. Under this regime, if performance were driven purely by node-specific memorization, we would expect all models pretrained on the same genes to behave similarly, which is not what we observe—GREmLN shows clear gains over competing encoders on held-out graphs.
>
> We will clarify this protocol and its implications in the revision, and explicitly discuss the distinction between node memorization and cross-graph generalization in the context of single-cell data. We are happy to continue the discussion on both this point and other concerns.

---

### Official Review · Reviewer_KWcx · 2025-10-29

**Soundness:** 3
**Presentation:** 3
**Contribution:** 2
**Rating:** 4
**Confidence:** 3

**Summary:**

In this study, the GREmLN (Gene Regulatory Embedding-based Large Neural model), a foundation model that leverages graph signal processing to embed gene token graph structure was proposed for single cell based gene embedding. The evaluation results showed improved cell type annotation accuracy.

**Strengths:**

A new GREmLN (Gene Regulatory Embedding-based Large Neural model) model, by leveraging (signaling, regulatory, protein interaction) graph signal processing to embed gene token graph structure for single cell based gene embedding.

The evaluation results showed improved cell type annotation accuracy (the improvement is limited).

**Weaknesses:**

The theoretical contribution is kind of limited considering that signaling graphs have been incorporated into graph AI models for single cell data analysis.
The improvement of performance compared with existing models is limited.

**Questions:**

It is important to have graphical illustrations to show the data analysis pipelines and model architectures.
It is unclear how many cells were used to train the proposed model.

---

> ### Author Response · Authors · 2025-11-21
> **Rebuttal**
>
> We thank the reviewer for their comments. Here are our responses:
>
> 1. We thank the reviewer for this comment. We agree that the general principle of propagating signal over graphs is well established in the GNN literature; however, our contribution lies in how this principle is instantiated in a transformer-style attention mechanism on large, sparse biological networks. Specifically, we apply a diffusion operator defined on a molecular interaction graph directly to the query representations, while leaving keys and values unchanged. This design has two technical consequences: (i) it enables us to lift signal propagation to the scale of genome-wide, highly sparse gene/protein graphs, using an efficient Chebyshev approximation to implement diffusion without explicit eigendecomposition, taking advantage of graph sparsity and size; and (ii) by transforming only the query and not the key/value tensors, the model injects an explicitly asymmetric graph-informed pattern into the attention scores, rather than relying on symmetric adjacency-based biases. To our knowledge, this combination is novel in the context of single-cell foundation models.
>
> 2. We thank the reviewer for spotting this. The model was pre-trained on 11M healthy human single-cell profiles from the CellxGene corpus. We will add illustrative data processing and model architecture diagrams as suggested by the reviewer.

---

### Official Review · Reviewer_XLky · 2025-10-31

**Soundness:** 2
**Presentation:** 1
**Contribution:** 3
**Rating:** 4
**Confidence:** 3

**Summary:**

GREmLN is a novel single-cell transcriptomics foundation model that resolves the orderless gene feature problem by directly embedding gene interaction graph structure into the Transformer via a graph diffusion kernel, yielding biologically meaningful and high-performing gene embeddings.

**Strengths:**

1. The paper is well-structured and easy to follow, with clear motivation and comprehensive explanations of each component in the proposed method.
2. GREmLN effectively captures long-range dependencies between genes by injecting the structural information into  Transformer's self-attention calculation.
3. This graph-integrated approach provides a powerful biological inductive bias for single-cell transcriptomics data, enabling the model to learn gene embeddings with higher biological significance and achieve outstanding performance in various tasks.
4. The paper provides the codebase for the baseline methods used and detailed hyperparameter specifications for the proposed method, which greatly facilitates reproducibility.

**Weaknesses:**

1. The motivation is questionable. In the Introduction, the authors state that “These formulations overlook the fact that gene products in the cell, as represented in scRNA-seq profiles, lack inherent sequential order or positional semantics,” where “These formulations” refers to methods such as scGPT [1]. However, in fact, scGPT removes causal masking and replaces positional encoding with gene name–based embeddings. Therefore, previous methods (at least scGPT) have already avoided the sequential design inherent to Transformers.  \
[1] Cui, Haotian, et al. "scGPT: toward building a foundation model for single-cell multi-omics using generative AI." Nature methods 21.8 (2024): 1470-1480.

2. The paper lacks an analysis of computational resource requirements. It proposes using a graph structure to provide relational information between genes for the Transformer. Given that the number of genes can be extremely large, this approach may demand substantial computational resources; however, the authors did not provide any analysis regarding these requirements.

2. The paper contains numerous formatting concerns: \
* The titles of the tables should appear above the tables, but in the paper, all table titles appear below them.
* The formatting of equations is inconsistent. Most equations lack numbering, while only some equations in Section 4.1 include equation numbers.
* The citation format seems incorrect. Except in the Introduction, all citations use the \citet{} format.

3. The paper lacks a discussion of the limitations of the proposed method.

4. The paper lacks a description of how LLMs are used.

**Questions:**

1. Given the method’s reliance on the gene relationship graph structure, could the authors provide an analysis of its computational resource requirements?

2. Could the authors provide a detailed discussion of the respective advantages of scGPT and the proposed method?

---

> ### Author Response · Authors · 2025-11-21
> **Rebuttal**
>
> We thank the reviewer for their comments. Here are our responses:
>
> 1. We thank the reviewer for pointing out this miscommunication. The intended meaning behind this phrase was to illustrate that the other models don't provide any meaningful biological context to the gene expression signatures. Although scGPT is trained to be gene-order-invariant, it simply adapts to a shortcoming of the scRNA-seq data, whereas we enrich the data with the relative positional encoding provided by the graph topology.
>
> 2. In terms of the computational overhead: to streamline pre-training, the fixed kernel gram matrices were precomputed during data pre-processing.Thus, the kernel gram matrix phi_L is loaded per sample and is used to transform the learned query Q, phi_L(Q). This leads to significant pre-training efficiency gains. Ultimately, GREmLN finishes 1 epoch in 11 hours versus 3 hours for a same-sized un-diffused transformer model (excluding validation steps) on a single H100 GPU. Precomputing the kernel graph matrices also allows for efficient scaling as the same gram matrix is used in every transformer layer.
>
> 3. We thank the reviewer for pointing out formatting issues. We will fix those in the updated version of the manuscripts, including the limitation discussion and LLM usage declaration.
>
> 4. Detailed comparison with scGPT: scGPT is designed to be a large, general-purpose single-cell foundation model trained on a very broad corpora with task-agnostic generative objectives. Its main strengths are (i) broad applicability across many downstream tasks (batch correction, annotation, imputation, etc.), (ii) robustness on heterogeneous datasets, and (iii) ease of use in settings where no high-quality prior graph is available. However, we see that scGPT does not produce a meaningful gene embedding(see masked edge prediction task on human immune cells), and is potentially overfitted to reconstruct gene expression. By contrast, GREmLN is a graph aware model that explicitly incorporates molecular interaction structure into the attention mechanism. Importantly, GREmLN does not require a curated gene regulatory network; it can take any gene- or protein-level graph as input and convert it into a diffusion-kernel prior over tokens. This yields (i) a principled solution to the lack of a natural gene order, (ii) strong inductive biases toward biologically plausible long-range dependencies, and (iii) a more direct handle on tasks that requires meaningful gene representations, such as reverse perturbation or network-level reasoning. In settings where such graphs are available and the scientific questions on gene level are important, GREmLN is therefore more targeted and sample-efficient, whereas scGPT remains preferable as a broadly applicable, task-agnostic backbone

---

> > ### Comment · Reviewer_XLky · 2025-11-25
> >
> > I greatly appreciate the authors’ detailed response and the effort they have put in. Most of my concerns have been addressed. However, I am still curious about the following questions and concerns.
> >
> > 1. Excessive training time overhead. The authors mention in their response that the training time of GREmLN is 3.67 times that of a same-sized undiffused transformer model (11 hours vs. 3 hours), introducing an additional 267% computational cost. However, the results in the paper indicate that this extra computational overhead does not lead to significant performance gains.
> >
> > 2. In Section 4.4, what are the detailed settings for Vanilla (ablation)?
> >
> > 3. The performance of GREmLN heavily depends on the correctness of the input graph structure. This limits the applicability of GREmLN as a foundational model.
> >
> > 4. The authors mention in Appendix A.5 that the model was trained for only one epoch. For high-dimensional, highly sparse, and noisy data types like single-cell data, a single pass is unlikely to sufficiently embed general biological knowledge into the model weights. Especially with large-scale datasets, multiple epochs are typically required for convergence.This strongly suggests that the model is underfitting and has not adequately learned the general biological patterns present in the data.

---

> > > ### Author Response · Authors · 2025-11-26
> > > **RE: Reviewer Comments**
> > >
> > > We thank the reviewer for engaging in active discussion. Here’s our responses to their concerns:
> > >
> > > 1. Excessive training time overhead. This is indeed an important question so we appreciate the reviewer bringing this up. GREmLN uses 10.3M parameters.Thus, while the diffusion kernel attention increases per-epoch wall-clock time relative to a same-sized vanilla transformer, the total compute footprint is still modest compared to existing scRNA foundation models. Additionally, to investigate performance gain fairly, we need to isolate the effect of the additional computation and compare it to an identical model backbone. Our ablation results suggest non-trivial gains on structure-sensitive and OOD tasks: for graph reconstruction on Human Immune, AUROC improves from 0.683 to 0.957; for zero-shot cell type annotation on held-out non-immune tissues, macro F1 increases from 0.816 to 0.937; and for reverse perturbation prediction, GREmLN surpasses scFoundation, Geneformer, and a linear baseline, with fine-tuning improving accuracy from 0.281 to 0.418.
> > >
> > > 2. Ablation Setting. We apologise for not including this and appreciate the reviewer pointing it out. The Vanilla ablation uses the same backbone as GREmLN: same embedding scheme (gene ID + rank embeddings), model dimension (512), number of transformer blocks (3), number of heads per block (8), activation (GeLU), and parameter count (≈10.3M).
> > > The only change is the removal of the graph diffusion kernel transformation of the queries. In the Vanilla setting, queries, keys, and values are computed as in a standard multi-head self-attention layer. No Laplacian, diffusion kernel, or Chebyshev approximation is used; attention scores are simple dot products. The Vanilla model is trained on the same pretraining corpus, with identical tokenization and masking scheme and the same masked modeling objective. We use the same optimizer (AdamW), batch size (64), learning rate schedule (warm-up then cosine annealing), and number of pretraining steps (one full pass over the corpus).
> > >
> > >
> > > 3. Dependency on graph structure. We thank the reviewer for pointing this out. Since we treat graphs as inductive bias, the model performance does depend on graph quality. However, we would like to point out that GREmLN is a graph-aware family of foundation models rather than one tied to a single graph type or inference pipeline. We demonstrate that both GRNs and PPIs can serve as input graphs, and the PPI network we use consists solely of biologically validated interactions without cell type– or dataset-specific tailoring (Appendix A.9), which we view as a broadly applicable inductive bias. The HINT PPI network, in particular, aggregates PPIs from multiple databases with automated and manual QC to reduce false positives. This mitigates the concern that we rely solely on a single noisy inferred GRN. Additionally, the fact that GREmLN generalizes to held-out cancer-infiltrating myeloid graphs and zero-shot non-immune cell types—where regulatory structure is known to be largely rewired—suggests that the model learns stable, reusable patterns rather than overfitting to any single GRN instance.
> > >
> > >
> > > 4. Potential Underfitting. We appreciate this concern and agree that pretraining schedule is a key design choice. The “one epoch” reported in Appendix A.5 corresponds to a full pass over a large CELLxGENE-derived corpus after stringent QC, quantization, and GRN or PPI construction. In practice, this involves a large number of gradient steps and exposure to a diverse set of tissues and cell types, rather than a small dataset that would almost surely underfit. GREmLN also operates at a substantially smaller parameter scale than existing foundation models, so we need less training time for the model to properly fit. Additionally, benchmarking results and gene expression reconstruction showcases that the model is fitted properly to understand transcriptomics landscapes. Given the size of the training corpus and our focus on exploring a new architectural inductive bias (graph diffusion kernel attention), we chose a single-epoch schedule to stay within reasonable compute limits (2× H100 GPUs) for this initial study. We agree that a systematic exploration of longer pretraining schedules (multiple epochs, larger models) is an important next step and will explicitly acknowledge this limitation and future direction in the Discussion. We are also actively training larger models with longer epochs and more data, which we will include in our final paper if results are available.
> > >
> > >
> > > We hope these clarifications address the reviewer’s concerns and we will incorporate all of the above points and additional details into Sections 4.4, A.2, A.4, A.5, and the Discussion in the revised manuscript. We are happy to discuss any further questions the reviewer has.

---

### Official Review · Reviewer_LeQb · 2025-11-01

**Soundness:** 2
**Presentation:** 3
**Contribution:** 3
**Rating:** 4
**Confidence:** 4

**Summary:**

This paper introduces GREmLN, a foundation model for single-cell transcriptomics designed to address the challenge that scRNA-seq data lacks the inherent sequential order exploited by standard transformers . The core contribution is a novel "Graph Diffusion Kernel Attention" mechanism . This method leverages graph signal processing to embed the structure of molecular interaction graphs, such as gene regulatory networks (GRNs) or protein-protein interaction (PPI) networks , directly into the attention mechanism. By applying a diffusion kernel derived from the graph Laplacian to the query embeddings , the model aims to produce biologically informed, single-cell-specific gene embeddings that capture long-range dependencies . The authors demonstrate GREmLN's effectiveness on several downstream tasks, including cell type annotation, graph structure understanding, and reverse perturbation prediction, claiming superior performance and parameter efficiency compared to existing baselines .

**Strengths:**

1. The paper tackles the significant and well-known challenge of meaningfully incorporating biological prior knowledge (in the form of molecular interaction graphs) into foundation models for scRNA-seq data, which is inherently orderless .
   2. The proposed method, which uses a graph diffusion kernel to structure the attention mechanism , is a technically novel to encoding this complex structural information.

**Weaknesses:**

1. **Overstated Novelty of the Problem Formulation:** The introduction's framing of the "orderless gene" problem seems to overlook how prior work has actively addressed this. The paper suggests this challenge remains largely unsolved . However, several baseline models mentioned in the paper have proposed their own explicit solutions. For example, scGPT and scFoundation utilize specific tokenization and/or attention strategies to handle the set-like nature of genes, while Geneformer and Cell2Sentence employ specific gene sorting methods before feeding the data into a transformer. The paper's contribution would be stronger if it were more precisely contextualized against these *existing* solutions rather than presenting the problem as unaddressed.
   2. **Marginal Performance Gains and Unclear Efficiency Trade-offs:** The performance improvement on the cell type annotation task, particularly when compared to scGPT, is marginal (e.g., 0.937 vs. 0.924 accuracy in Figure 1 ; 0.939 vs. 0.924 Macro F1 in Table 1 ). While the paper rightly notes GREmLN's improved parameter efficiency (10.3M parameters ), it fails to provide a complete picture of the *computational* trade-offs. The Graph Diffusion Kernel Attention, with its reliance on Laplacian operations and Chebyshev approximations , likely introduces non-trivial computational overhead during training and/or inference. The evaluation is missing a direct comparison of training times and inference latencies, making it difficult to assess the method's true efficiency and scalability.
   3. **Limited Perturbation Benchmarking:** The experimental validation for perturbation analysis in Section 4.5 is incomplete.
      - First, the paper only evaluates *reverse* perturbation prediction (identifying the perturbation from a perturbed profile). A standard and critical benchmark for many scRNA foundation models is *forward* gene perturbation prediction (i.e., *in silico* prediction of gene expression after a perturbation), which is not included.
      - Second, scGPT is notably absent from the baselines in the reverse perturbation task (Table 3 ). Given that scGPT is a primary competitor in all other tasks and has been benchmarked on perturbation tasks, its exclusion from this comparison is a significant omission.

**Questions:**

1. Could the authors please clarify the novelty of their approach to handling orderless genes in light of prior work? Specifically, how does GREmLN's graph-based embedding mechanism fundamentally differ from, and improve upon, the tokenization strategies of scGPT/scFoundation or the explicit gene-ordering methods used by models like Geneformer, which also aim to address this exact problem?
2. Regarding the modest performance gains in cell type annotation (Table 1 ), what is the practical computational cost (e.g., training time, inference latency) of the Graph Diffusion Kernel Attention compared to the standard attention mechanisms in scGPT, Geneformer, and scFoundation? A direct comparison of these efficiency metrics is needed to fully evaluate the model's scalability and utility.
3. Regarding the perturbation experiments (Section 4.5 ):
   - a) Why was scGPT omitted as a baseline in the reverse perturbation prediction task (Table 3 )?
   - b) Why did the authors opt to only evaluate reverse perturbation prediction, and not the more common task of in-silicon single-cell gene perturbation prediction?

---

> ### Author Response · Authors · 2025-11-21
> **Rebuttal**
>
> We thank the reviewers for their insightful comments. Here’s our response to address the reviewer’s concerns:
>
> 1. Prior work models, such as Geneformer, enforce a gene sequence; however, this is not biologically meaningful. Geneformer order genes by expression level, which does not reflect underlying biology - much like ordering the words in a sentence by their length does not capture linguistic meaning, since high expression does not necessarily indicate biological functions. As for scGPT and scFoundation, these are designed to be robust against unordered gene sequences. scGPT explicitly randomized the input order and uses no positional encodings. scFoundation uses set-compatible Transformer attention for permutation invariant performance. Although this addresses the set-like nature of genes, we argue that the additional context provided by a relative positional encoding of genes through the lens of regulatory networks provides a biologically meaningful structural prior that better informs transformer attention, in a novel implementation.
>
> 2. We agree with the reviewer that the performance improvement on the cell type annotation task is marginal. We would like to emphasise that the benchmarking performances on this task are top-saturated, thereby limiting the extent of potential improvement. Additionally, we would like to bring up the following: a) We outperform other models more significantly in the zero shot setting, which is more common in real application scenarios. b) This is a common benchmark in single-cell foundation model evaluation that we feel is best included, although we seek to further demonstrate biological relevance with our reverse perturbation prediction benchmark, which is a harder task methodology wise too(see 4 for more detail).
>
> 3.To streamline pre-training, the fixed kernel gram matrices were precomputed during data pre-processing. Thus, the kernel gram matrix phi_L is loaded per sample and is used to transform the learned query Q, phi_L(Q). This leads to significant pre-training efficiency gains (~5x speed-up). Ultimately, GREmLN completes 1 epoch in 11 hours (excluding validation steps) on a single H100 GPU, while a same-sized un-diffused transformer model takes 3 hours. Precomputing the kernel graph matrices also allows for efficient scaling as the same gram matrix is used in every transformer layer.
>
> 4. We thank the reviewers for raising this point and apologize for not sufficiently motivating our experimental design. While the forward perturbation prediction task has indeed been extensively studied, we would like to emphasize two clarifications:
> a) Due to the nature of genetic perturbation data, commonly used evaluation metrics for forward prediction are substantially flawed and can compromise the fairness and interpretability of benchmarking. For example, prior work has shown that different, reasonable metric formulations can lead to seemingly conflicting conclusions about model performance on the same datasets[1, 2]. Providing a definitive solution to this metric-design problem is beyond the scope of the present work. In contrast, the reverse perturbation setting admits standardized and well-understood classification metrics. b) The reverse perturbation task is both more biologically aligned with typical use cases and a more direct probe of representation quality. Biologically, this setting matches concrete applications such as mechanism-of-action inference, target deconvolution, and identification of functionally similar interventions: given an observed cell state, we want to know which perturbation(s) could have produced it. Methodologically, this task places stricter demands on the learned embeddings. A model must organize the representation space so that (i) perturbations inducing similar transcriptional responses map to nearby regions, and (ii) biologically distinct perturbations remain separable, ideally allowing even a simple classifier trained on frozen embeddings to recover perturbation labels with high accuracy. These tasks are substantially more challenging on genetic perturbation data, where effect sizes are subtle and perturbation-induced transcriptional responses are highly homogeneous. In this regime, different perturbations often produce only small, noisy shifts in expression, leading to tightly overlapping clusters that are difficult to separate in embedding space.
>
> 5. We thank the reviewer for point this out. We will add comparison to scGPT by the final deadline on this task. The scGPT analysis did not finish running by the submission deadline
>
>
> [1] Miller, Henry E., et. al. . 2025. “Deep Learning-Based Genetic Perturbation Models Do Outperform Uninformative Baselines on Well-Calibrated Metrics.” bioRxiv. https://doi.org/10.1101/2025.10.20.683304.
> [2] Ahlmann-Eltze, Constantin, Wolfgang Huber, and Simon Anders. 2025. “Deep-Learning-Based Gene Perturbation Effect Prediction Does Not yet Outperform Simple Linear Baselines.” Nature Methods 22 (8): 1657–61.

---

### Author Response · Authors · 2025-12-03
**Final Remarks**

As the rebuttal phase come to an end, we would like to thank the reviewers for their thoughtful and constructive feedback. In the rebuttal phase, we substantially clarified the novelty and positioning of GREmLN relative to scGPT, scFoundation, Geneformer and related work, emphasizing how graph diffusion kernel attention provides a distinct, biologically grounded solution to the orderless-gene problem. We added a detailed analysis of computational cost and ablation settings (including the Vanilla backbone), clarified our pretraining schedule and dataset scale, and expanded the perturbation section with a clearer motivation for reverse perturbation, improved discussion of evaluation metrics, and the inclusion of scGPT as a baseline where feasible. We further elaborated the role and robustness of the input graphs (GRNs vs. PPIs), the Chebyshev/kernel computation, Bayesian graph integration, and the edge-prediction evaluation protocol, addressing concerns around dependence on graph quality and potential evaluation bias. We hope these rebuttals address the reviewers’ concerns and significantly strengthen the manuscript. The final manuscript will incorporate all reviewer suggestions caveats and additional results presented during the discussion phase, such as extended benchmarking with scGPT and model architecture/data preprocessing schematics. Overall, we believe that GREmLN’s conceptual framework provides a critical step toward embedding mechanistic information into omics foundation models for downstream biological applications.

---

### Meta-Review · Area_Chair_sEiq · 2026-01-02

**Summary:**

GREmLN introduces a single-cell transcriptomics foundation model that injects molecular interaction graphs into Transformer attention via diffusion-based query smoothing to learn biologically informed gene/cell representations for downstream analysis.

Reviewers highlight strengths including the timely motivation (using biological priors for orderless gene tokens), a technically coherent graph-aware attention design with scalability considerations, and generally strong empirical results with supportive ablations.  Meanwhile, they also point out some weakness: (i) the novelty/positioning is not sufficiently distinguished from prior scRNA foundation models, (ii) the practical cost is under-specified and appears substantially higher than a same-sized undiffused Transformer baseline, and (iii) the approach’s reliability depends heavily on graph construction/integration choices (GRN quality; test-time “Bayesian Graph Integration” via an auxiliary classifier), with limited robustness evidence.

Rebuttal clarifies some scope points but does not adequately resolve the central issues above.  Given these unresolved concerns that directly impact the credibility and practicality of the claims, I recommend rejection.

**Reviewer Concerns:**

See summary.

**Reviewer Scores:**

Overall, the reviewers’ scores are mixed to negative, with consistent concerns about novelty, evaluation validity, robustness, and practical efficiency outweighing the acknowledged strengths.

---

### Decision · Program_Chairs · 2026-01-26

Reject